# Improving Drug–Drug Interaction Extraction with Gaussian Noise

**DOI:** 10.3390/pharmaceutics15071823

**Published:** 2023-06-26

**Authors:** Marco Molina, Cristina Jiménez, Carlos Montenegro

**Affiliations:** Department of Informatics and Computer Science, Faculty of Systems Engineering, Escuela Politécnica Nacional, Av. Ladron de Guevara E11-25, Quito 170525, Ecuador; carlos.montenegro@epn.edu.ec

**Keywords:** drug–drug interaction, deep learning, BioBERT, relation extraction, DDIExtraction2013 challenge

## Abstract

Drug–Drug Interactions (DDIs) produce essential and valuable insights for healthcare professionals, since they provide data on the impact of concurrent administration of medications to patients during therapy. In that sense, some relevant works, related to the DDIExtraction2013 Challenge, are available in the current technical literature. This study aims to improve previous results, using two models, where a Gaussian noise layer is added to achieve better DDI relationship extraction. (1) A Piecewise Convolutional Neural Network (PW-CNN) model is used to capture relationships among pharmacological entities described in biomedical databases. Additionally, the model incorporates multichannel words to enrich a person’s vocabulary and reduce unfamiliar words. (2) The model uses the pre-trained BERT language model to classify relationships, while also integrating data from the target entities. After identifying the target entities, the model transfers the relevant information through the pre-trained architecture and integrates the encoded data for both entities. The results of the experiment show an improved performance, with respect to previous models.

## 1. Introduction

Drug–Drug Interactions (DDIs) are changes in a drug’s effects due to the ingestion of another drug or drugs by a patient in the same period of time. Such interactions can not only cause unexpected and dangerous side effects, but also an increase in healthcare costs [1]. As a result, in recent years, the detection and extraction of DDIs has become an important research area for patient safety. One of the field’s main research goals is to aid physicians in gathering knowledge about DDIs by developing new technologies that can make this search effortless. Commonly, physicians obtain information regarding DDIs from two main sources of data, i.e., from either analyzing vast amounts of biomedical papers or using biomedical databases to query DDIs [2,3]. However, reading lots of papers or manually querying professional databases are not efficient ways to gather knowledge about DDIs since both are tedious and time-consuming tasks.

In the field of natural language processing (NLP), extracting DDIs is a typical task for relation extraction. As a matter of fact, some of the most important results in the biomedical field have been achieved by using the RE model; the correct semantic and structured relation between a pair of drug entities can be identified. For instance, RE has been applied in analyses of interactions between proteins [4], disease–treatment relationships [5] and drug–drug interactions [3].

Numerous early methods for extracting DDIs from biomedical literature can be broadly categorized into two groups: traditional machine learning and deep learning. The traditional approaches used extraction rules and traditional machine learning relies on feature engineering, which depends on the domain knowledge to characterize the problem mathematically. In contrast, advances in deep learning alleviate the feature-learning problem, since the most suitable set of characteristics are automatically extracted from data. As a consequence, Convolutional Neural Networks (CNNs), a deep learning architecture, have been able to extract relevant features from data with satisfactory results in DDIs. Moreover CNNs have improved performance in DDI extraction compared to rule-based approaches and traditional machine learning [6,7,8]. In spite of the performance achieved by CNNs in DDI extraction, significant improvements have been observed with the use of pre-trained language models (PLMs), such as BERT over traditional Deep Neural Networks (DNNs) in Relation Classification (RC) tasks [9,10,11].

Data augmentation techniques can help improve the robustness of the aforementioned models. These techniques utilize several algorithms that can generate synthetic data from existing datasets [12]. Consequently, the model’s predictions turn out to be invariant to the small changes applied to data through the use of data augmentation. What is more, creating synthetic data enables the model to learn from examples that are not present in the original dataset and to identify complex patterns that may be hard to discern otherwise [13]. In this work, data augmentation is used by injecting Gaussian noise just before the Softmax classifier.

In order to provide a framework for evaluating different extraction techniques, the DDIExtraction2013 corpus has become an important standard for researchers working on the recognition of active pharmacological substances and related relation-extraction tasks, specifically for DDI detection and classification [14]. In the literature, the most frequent performance metric used to report experimental results is F1-Score. The DDIExtraction2013 Task challenge was proposed for recognizing and classifying drug names and for extracting and classifying their interactions. The DDI detection and classification task attracted a strong response from the research community, with 14 teams submitting 38 runs in total. The team with the best performance achieved an F1-score of 0.715 for drug name recognition and classification and 0.651 for interaction extraction and classification. The most efficient model proposed in this challenge [3] is based on Support Vector Machine (SVM) with multiphase kernel [15]. The second best result was achieved via a model that used two SVM linear classifiers with special characteristics such as precise word order and syntactic constructions [16]. Other proposed models used linear kernel SVM, and relied on different features, including trigger words and kernel parsing of the DDI tree [17,18]. Subsequent to this challenge, new approaches appeared. Deep learning neural networks, such as Convolutional Neural Networks (CNNs) and Long Short-Term Memory Networks (LSTM), have been used for DDI extraction. These approaches improved the designed network and its performance [19]. Recent studies suggest that developments in natural language processing (NLP) have opened up new possibilities for biomedical NLP applications. Language models that have been trained or fine-tuned with domain-specific corpora have demonstrated superior performance compared to general models. However, biomedical NLP has been constrained in terms of both corpora and tasks thus far [10].

This research project proposes the use of data augmentation techniques; we will characterize two state-of-the-art models that exhibit good performance. The strategy to be used is to inject perturbations using Gaussian noise. Much research in recent years has focused on the DDIExtraction2013 challenge with numerous strategies to improve performance relative to previous models. There remains a need for an efficient model capable of obtaining results with a high degree of confidence.

In order to present our findings, we have organized this paper into the following sections: The Introduction reviews some definitions, techniques and challenges of DDI extraction and RE concepts in NLP. The methodology and architecture of the models used in this research are presented in Section 2. Experimental results, models’ parameters and models’ performance are presented in Section 3. Section 4 presents a discussion on the following items: comparison of the performance of the most advanced methods with our models and analysis of errors. Conclusions of this research work, as well as some suggestions for future work, are presented in Section 5. Finally, the references used in the literature review of this work are shown.

## 2. Materials and Methods

In this work, we propose two ER models: (1) One based on a PCNN convolutional neural network, with a multichannel approach [20]. (2) Another based on a pre-trained BERT language model and which incorporates information from two target entities using bidirectional Transformers. Our models are evaluated with the DDIExtraction2013 corpus; the results are reported, compared and analyzed.

In this section, we describe the proposed models, based on the aspects that refer to the use of Gaussian noise.

### 2.1. Dataset

The proposed models were trained and evaluated with the DDIExtraction2013 corpus, which can be downloaded from the following link: (https://www.cs.york.ac.uk/semeval-2013/task9/ (accessed on 20 December 2020)).

The DDIExtraction2013 challenge provides a reference dataset, the DDI Corpus, which is a collection of manually annotated texts. This corpus is a good standard for training and evaluating supervised machine learning algorithms to extract DDIs from biomedical literature.

Figure 1 shows an example of corpus content, where the text (“Tetracycline, a bacteriostatic antibiotic, may antagonize …”) can be seen, along with some pharmacological entities, with their identification (e0: Tetracycline; e1: bacteriostatic antibiotic; e2: penicillin) and the relationship class “effect”.

The DDIExtraction2013 corpus is one of the most important datasets; the performance of various DDI extraction methods is evaluated and compared using this corpus, which consists of 730 documents representing DDIs from the DrugBank database and 175 abstracts on DDI subjects from MedLine database. The corpus was divided into training and test datasets to evaluate the participating systems. The training dataset was randomly taken from about 77% of corpus files and the remainder were used as the test dataset. The DDI 2013 corpus contains five DDI types: Advice, Effect, Mechanism, Int and false.

Mechanism. When the text describes the process through which the interaction takes place.Effect. When the text describes the outcome of drug interactions.Advice. When the text describes recommendations or adviceInt. When a DDI is provided without any information.False. If the influence of one drug on another is not described in the text, the DDI belongs to the false class.

Identifying potential Drug–Drug Interactions (DDIs) in text involves recognizing whether a pair of drugs mentioned in a given sentence interact with each other and, if so, in what way. Various interaction types between a pair of drugs are shown in Table 1 by means of examples.

The corpus is divided into classes for test sets and a training set; this is shown in Table 2.

An evaluation process was conducted to determine the optimal model by selecting the available parameters independently in a validation set to obtain the highest values. Considering that the DDI Corpus is exclusively separated into training and test datasets, 10% of the instances were carefully selected from the training dataset, at the sentence level, forming the validation set, which was used for each experiment to fine-tune the hyper-parameters for the model’s architecture.

### 2.2. Proposed Model No.1

Figure 2 shows a model’s architecture that includes seven parts: Multichannel integration, Word embedding, Convolution, Max pooling, Attention mechanism, Gaussian noise and Softmax classifier.

#### 2.2.1. Architecture

The model uses a multichannel convolutional network (MCCNN). The channel concept is inspired by three-channel RGB image processing [21], which means that the inclusion of different words represents different channels and different aspects of the input words. Multiple versions of word embeddings are integrated into the model, as presented by Quan et al. [18], which improve the representation of input words that belong to sentences containing pairs of drugs.

Additionally, RE for DDI is addressed by applying a feature-extraction model. The PW-CNN model or CNN part-based model, suggested by Zeng et al. [19], is applied to overcome the limitations during the max-pooling process, where the hidden layer size increases very quickly. Thus the model integrates five versions of word embeddings, for better representation of words.

#### 2.2.2. Word Embeddings

The concept of word embeddings may be visualized as a vector of $R^{k}$, representing a single word depending on the context in which it appears. The words of a vocabulary are thus mapped to a point in vector space, as shown in Figure 3.

The neural network inputs, shown in Figure 2, consist of sentences that contain pairs of drugs. This approach is based on a simple principle: Words that share similar contextual patterns tend to have similar meanings. This idea forms the basis of word-embedding techniques, which analyze distributions of words in an unlabeled corpora such as those found in PubMed, PMC, MedLine and Wikipedia. To produce a comprehensive set of embeddings, we generated five different versions using these sources. The training corpora used are listed in Table 3. The first four word embeddings are released on [22] and the fifth one is trained via CBOW on the MedLine corpus (http://www.nlm.nih.gov/databases/journal.html (accessed on 20 December 2020)).

Unidentified word problems often arise due to the use of unfamiliar biological terms. To compensate for this problem, the biological corpus PubMed, PMC and common corpora such as MedLine and Wikipedia are used. Word embeddings are used for each input channel. As mentioned above, word embeddings are distributed representations of words where a vector of $R^{k}$ is assigned to each word within a text. Recent studies have shown good performance in capturing semantic and syntactic information from sentences [23]. The use of embedding words, which have been previously trained, has become a valuable contribution with respect to improving NLP tasks [6]. The word-embedding method of learning is unsupervised, and exploits word-matching structure in untagged text. Researchers have proposed various training word-embedding methods [23,24]. The key position is added as an input value of the model. This embedding position determines the relative distance between the two entities and the remaining words in a sentence. This process makes it possible for the relative location of the two entities to be specified. As an example, in the sentence of Figure 4, the relative distances of e1 and e2 with respect to the word “decrease” are established.

#### 2.2.3. Convolution

Convolutional neural networks have been applied successfully in the field of computer vision; however, in recent years, these models have also been applied to NLP. CNNs have the ability to extract salient features of n-grams, from an input sentence, to create a semantic representation of it. The works of [24,25,26] are samples of the proliferation of networks based on CNN in fields other than image recognition. The following describes the operation of a simple CNN-based sentence-modeling network. The feature-extraction process is performed by the convolution layer.

Let $x_{i}\in R^{k}$ denote the *k*-dimensional word vector associated with the *i*-th word in the sentence. A sentence of length *n* is represented as in Kim [26]:(1)$$x_{1:n}=x_{1}⊕x_{2}⊕\ldots ⊕x_{n}$$
where ⊕ is the concatenation operator. In general $x_{\left(i:i+j\right)}$ refers to the linking of words $x_{i},x_{\left(i+1\right)},\ldots ,x_{\left(i+j\right)}$. In a convolution operation, $w\in R^{hk}$, a filter is applied to a window of *h* words to generate a new characteristic. For example, a characteristic $c_{i}$ is produced from a window of words $x_{\left(i:i+h-1\right)}$ according to the following equation:(2)$$c_{i}=f\left(w*x_{\left(i:i+h-1\right)}+b\right)$$
where b∈R is a bias component and *f* is a nonlinear function, including the hyperbolic tangent. This filter is applied to every possible window of words in the sentence $x_{\left(1:h\right)},x_{\left(2:h+1\right)},\ldots ,x_{\left(n-h+1:n\right)}$ to generate the feature map, Equation (Equation 3):(3)$$c=c_{1},c_{2},\ldots ,c_{\left(n-h+1\right)}$$
where $c\in R^{\left(n-h+1\right)}$. Then a maximum overtime grouping operation is applied to the feature map [24] and the maximum value is taken, Equation (Equation 4):(4)$$\hat{c}=max\left[c\right]$$

The aim is to capture the characteristic associated with this specific filter. The goal is to extract the most significant feature with the highest value for each feature map. The model incorporates multiple filters (with different window sizes) to derive multiple functions. These characteristics form the penultimate layer and then are fed into a Softmax layer, the output of which is the probability distribution on the labels.

#### 2.2.4. Piecewise Max Pooling

The PW-CNN model was proposed by Zeng et al. [19]. This is a model that turns out to be suitable for the task of remotely supervised REs. The success of this process depends on extracting the correct structural features from the sentence containing the entity pair.

The output of a CNN network depends on the number of tokens in the sentence, a dependency that can be eliminated, if the max-pooling operation is applied.

If the sentence $c_{i}$ is supposed to contain two entities and is divided into three parts $c_{i}1;c_{i}2;c_{i}3$, each part will correspond to a context of the sentence:$c_{i1}$ contains the context of the sentence to the left of the first entity;$c_{i2}$ contains the context of the sentence between the two entities;$c_{i3}$ contains the context of the sentence to the right of the second entity.

Next, a maximum grouping is performed in each of the three contexts, as shown in Figure 5.

In this way, the location information of the entities is used to retain the structural characteristics of the sentence, after the max-pooling operation, Equation (Equation 5).
(5)$$pc_{ij}=max\left(c_{ij}\right);1\leq i\leq n,1\leq j\leq 3$$

The result of this operation is the concatenation of $pc_{i1};pc_{i2};pc_{i3}$ producing a fixed size output.

#### 2.2.5. Attention Mechanism

The attention mechanism strategy in a convolutional neural network (CNN) is used to improve the performance of sentence classification. In our model, the attention mechanism automatically captures contextual information and correlation between non-consecutive words without any external syntactic information; this is done giving more weight to the important words of the sentence. We carry out experiments on several datasets and show that the part-based multichannel convolutional model (MCPCNN) plus an attention mechanism (+Att) surpasses the basic MCPCNN and achieves good performance since they exploit the syntactic characteristics. Later on, we show more information.

#### 2.2.6. Gaussian Noise

As explained by Arslan et al. [27], Gaussian noise is a statistical approach noise with a probability density function equal to that of the normal distribution. This normal distribution is described as a continuous variable probability distribution where the data are clustered around a central value. The noise that occurs is due to random causes, but the most of random events follow a pattern. This is the main reason why in data science and machine learning Gaussian distribution is preferred; Gaussian random events are more frequently found in natural events. The graphical representation of this type of random event is the normal distribution curve, also called Gaussian bell.

As described in Papadaki [12], in this method, instead of generating new augmented sentences, perturbations are inserted into the word embeddings. For each word insertion, a Gaussian noise value is added according to the following equation:(6)$$\;w_{j}^{\prime }\;=w_{j}+x_{j}$$

$w_{j}$: the original word insertion.

$w_{j}^{\prime }$: the resulting word insertion.

$x_{j}$: the random noise value.

The noise vector values were tested from the truncated normal distribution with μ=0, σ=1 and with a range between 0 and 0.3

When the perturbation is set, the vector elements are taken randomly with a probability of 0.3, resulting in the Gaussian noise vector. The values are selected from a small range so that the resulting word embedding does not stray too far from the contextual word embedding space, i.e., the nearest neighbors are selected for new samples; see Figure 6. A characteristic of this method is that perturbations are inserted into all words, independent of their POS (part-of-speech) tag.

In the experiments, a layer of Gaussian noise was added just before activation of the Softmax classifier to simulate the effect of increasing data and avoid overfitting. These concepts have been used in the area of computer vision [28,29,30] and this approach is applied for this challenge.

#### 2.2.7. Softmax Output

Before classifying, the dropout technique avoids network overstress. For this, the elements of the vector *z* are randomly set to zero, with probability *p* in a Bernoulli distribution, to generate a reduced vector $z_{d}$. This vector feeds a fully connected Softmax layer with weights $W_{s}\;inR^{mk}$ to calculate the output prediction values for the classification, in accordance with Equation (Equation 7):(7)$$o=z_{d}W_{s}+d$$
where *d* is a bias term. In the dataset, there are k=5 classes (advice, effect, interaction, mechanism and no DDIs). At the time of testing, the vector *z* of a new instance is classified directly via the Softmax layer.

### 2.3. Proposed Model No. 2

BERT is a pre-training methodology for language representations that has been employed to produce models that can be downloaded and used free of charge by NLP researchers. By utilizing these models, one can extract high-quality language features from text data or fine-tune them for a specific task (classification, entity recognition, answering questions, etc.) with your own data to generate state-of-the-art predictions.

#### Architecture of Pre-Trained Model BERT

As proposed by [9], the pre-trained BERT model was applied with some modifications. This model is a multi-layer bidirectional Transformer encoder. The BERT model approach is to represent a pair of text sentences or one in a sequence of tokens. The input of each token is generated by summing the token, segment and position embedding. At the beginning of each sequence, the string [CLS] is used as the first token in the sequence. The final hidden state from the transformer encoder output associated with the first token is applied as the representation of a sentence for classification tasks. When there is a pair of text sentences, the string [SEP] is used as a separator of two sentences. Figure 7 illustrates the overview of the BERT architecture for DDI extraction. By adding special tokens at the start and end of the two entities in the sentence, the BERT module can generate target entity representations with ease. This is because the special tokens provide the necessary location information to the module, we introduce a special token #. We also attach [CLS] to the beginning of each sentence [10].

Task definitionIn this research, we emphasise relation representations.Let $x=[x_{0},\ldots ,x_{n}]$ be a set of tokens and (e1) and (e2) the entities established, as can be seen in Figure 7. ei∈x is denoted by the initial and final positions, $s_{i}=\left(e_{i}^{8},e_{i}^{e}\right)$.A relation statement is represented by a set of three components, known as a triple r=(x,e1,e2). The goal is to explore and describe a function $h_{r}=f_{\theta }\left(r\right)$ that maps the relation statement to a fixed-length vector $h_{r}\in R^{d}$ to represent the relation in *r*. It is noticeable that the two entities have partitioned the sentence into five parts, $e_{1}$ and $e_{2}$ as entity mentions and three contextual segments, represented as $c_{0},c_{1}$ and $c_{2}$.Entity positional encodingThe positional encoding gives information on the token position in the sequence and its relative distance to the other tokens in the input sequence *x*; see Figure 7. We will use the entity’s positional encoding 1 at the positions of e1, and 2 at the positions of e2, with the other tokens in the contextual sentence being given as 0. This entity positional encoding refers to an entity positional embedding module in the RC model and it is randomly operated and fine-tuned during BERT fine-tuning.Pooling layerIn this study, our focus is on selecting diverse pooling operations for different parts of the sentence. This includes all operations of pooling: average pooling, max pooling, self-attention pooling and start pooling, which is employed in the representation of the starting token.Output featuresTo choose accurate features for classification categories, some approaches are described: first, whether the two entity vectors should be applied as features; second, whether each contextual part (c0,c1,c2) should be included as features [10,11]. We analyze how the two entities interact with each other and with the contextual parts, considering that the interaction can be a dot product or absolute difference between two feature vectors.

## 3. Results

The following is the configuration of the models used in the experiments, as well as the results and discussion of the accuracy achieved.

### 3.1. Text Preprocessing—Model 1

In the preprocessing step, drug entities are anonymized, then sentences are tokenized, tokens are normalized and non-interacting pairs are filtered into selected DDIs, finally, the PW-CNN module is applied for DDI extraction.

The PW-CNN module is initialized with the generation of DDI instances from a pair of drug entities detected in the input sentence.

Given an entity phrase, there are $\left(c_{n}^{2}\right)$ instances that need DDI classification. In order to allow for generalization during learning and ensure that only one drug pair exists in each instance, we replace the two candidate entities with the symbols “$drug_{1}$” and “$drug_{2}$”; all other drug entities are substituted with “$drug_{n}$”. Please confirm if the italics is unnecessary and can be removed. The following highlights are the same

When processing a sentence that has *n* detected drug entities, the DDI instances are anonymized using $Drug_{1}$ and $Drug_{2}$ for the target drug pair and the other drugs are substituted with $Drug_{n}$. Equation (Equation 8) determines the number of detected instances.
(8)$$C_{\left(n,2\right)}=\frac{n(n-1)}{2}$$

For example, in the following sentence, three drug entities are detected: “Aminoglutethimide decreases the effect of coumarin and warfarin”. Applying Equation (Equation 8), C(3,2)=3(3−1)/2=3 DDI instances, as shown in Table 4.

#### Deletion of Negative Instances

In machine learning, unbalanced data affect model performance. Therefore, in several studies using the DDI 2013 corpus [31,32], to avoid performance degradation, two rules were implemented to filter negative instances that generate “false data”. These rules are described below:Eliminate any pair of drugs that refer to the same drug. This can be two drugs with the same name or synonyms.Filter drug pairs that share coordinated relationships. A coordinated relationship between two words is established by using a conjunction or a comma. This relationship often occurs between three or more drugs in negative instances.

### 3.2. Text Preprocessing—Model 2

In the preprocessing step, we use the preprocessed dataset from this link (https://github.com/arwhirang/DDI-recursive-NN (accessed on 15 February 2021)). We do not replace the name of the drug with ‘DRUG0’, ‘DRUG1’ or ‘DRUGN’.

In summary, to preprocess the data, we first add the [CLS] token at the start and the [SEP] token at the end of each input sentence.

Then the next steps follow.

Tokenization: split the sentence into tokens.Add the special [CLS] token at the start of the sentence.Add the special [SEP] token at the end of the sentence.Apply [PAD] tokens for sentence padding until the maximum length is reached.Map each token to its corresponding ID in the model.

### 3.3. Model 1: Experiments

In this work, a high-performance cluster was used, where the code is written in Python. In the implementation, the Keras libraries (https://keras.io/ (accessed on 15 February 2021)) and, as a back-end, TensorFlow (https://www.tensorflow.org/ (accessed on 15 February 2021)) were used. Five different word-embedding types were utilized—PMC, PubMed, PMC and PubMed, Wikipedia and PubMed, and MedLine (http://code.google.com/p/word2vec/ (accessed on 15 February 2021))—to train word representation input. Table 5 shows the parameters used in the experiments. The following models were implemented in this framework to make comparisons and improvements to the proposed model: BI-LSTM (Bidirectional Long Short-Term Memory), MCCNN, PW-CNN, MCPCNN; these models are discussed in greater detail later. The code and results obtained with Model No. 1 can be found on our GitHub repository (https://github.com/macjimenez/ModelMCPCNGaus (accessed on 20 March 2021)).

It is important to reiterate that we established a maximum sentence limit of 150 characters. When the text length is smaller, it is necessarily filled with 0 or else we trim the sentence to its maximum length before applying our model’s hyperparameters, which are listed below:We established a maximum sentence length of 150 characters for sentences that refer to DDI pairs.The number of hidden units in our model is 128.The batch size is equal to 64.The dropout rate is in the range 0.1 to 0.50.The context window size is [3 5 7 9].The learning rate is equal to 0.003.We utilized early stopping criteria over a period of 100 epochs to identify the top-performing model on the validation set.

### 3.4. Model 2: Experiments

The model is developed with Python 3.5, the Torch libraries (https://pytorch.org (accessed on 17 February 2021)) and scikit-learn and transformer scikit-learn. It already has a predefined configuration; no tests had to be carried out to find the most suitable configuration. The optimal parameters for this model can be seen in Table 6. The code and results obtained with Model No. 2 can be found on our GitHub repository (https://github.com/macjimenez/modelBERT (accessed on 20 March 2021)).

It is known from the literature that Relation Classification (RC) is an important NLP RE task. The principal methods for RC are based on CNN or RNN. As mentioned before, with its exceptional performance in multiple NLP classification applications, the pre-trained BERT model is a popular choice for many researchers. Our study presents a novel approach that utilizes the pre-trained BERT language model and incorporates target entity information to tackle the RC task. In this work, we must locate the target entities and transfer the information through the pre-trained architecture and incorporate the corresponding encoding of the two entities. Comparing the effect of pretrained language models using R-BERT architecture with different pretrained weights, we chose BioBERT v1.1 PubMed (See Table 7).

### 3.5. Evaluation of Information-Extraction Techniques Applied to DDIExtraction2013 Corpus

The F1-Score metric was used to evaluate the performance of the proposed models, which is the harmonic mean of precision and recall. This metric is used primarily when data are not balanced. The dataset in the DDIextraction2013 corpus is uneven because each data type is not uniform.

The architectures of BI-LSTM, PW-CNN [20], MCCNN [18], MCPCNN [20] and BERT (https://github.com/monologg/R-BERT (accessed on 10 August 2021)) were implemented.

In the experiments, the processed values are recorded (see Table 8, the items marked with an asterisk are the baseline models used in our study). The BI-LSTM model reached an F1-score of 0.6516; a classical architecture was implemented; more robust strategies that achieve higher F1-scores are applied to this model, such as in the work of Lamurias et al. [33] which uses ontologies and reaches an F1-score of 0.751; in the work of Zhou et al. [15], the authors use relative positions of the words and reach an F1-score of 0.73; the hierarchical model of Zhang et al. [34] uses attention mechanisms and the authors reach an F1-score 0.729.

In our experiments, the convolutional model by parts PW-CNN presented an F1-score of 0.6835. The MCCNN multichannel model presented an F1-score of 0.707. In the work of Quan et al. [18], the MCCNN model presented an F1-score of 0.702 on the same corpus; the registered values are very similar. The model proposed by Park et al. [20] MCPCNN, in our experiments, achieves values similar to those reported in the publication, F1-score 0.7174.

To enhance the F1-score of DDI extraction, the BERT model is refined by an average operation to obtain a vector representation for each of the two target entities. In accordance with the experiments carried out on the DDIExtraction2013 dataset, the F1-score of our model reaches 0.822, which is a better performance compared to the existing models such as CNN and LSTM.

Our Model No. 1 was implemented based on the work of Park et al. [20]; Gaussian noise was added, which is a data augmentation technique to simulate data increase, achieving an F1-score 0.7382.

Our Model No. 2 was implemented based on the work of Nguyen and Ho [35]; a Gaussian noise layer was added to this model. The same technique was used as for Model No.1, achieving an F1-score of 0.8438. The Gaussian noise layer is programmed because there is no such library in pytorch.

We can observe an improvement in performance (see Table 9); the MCPCNN Model No. 1 reaches an F1-score of 0.7174, which is the same value reported in the work of Park et al. [20]. By adding Gaussian noise and the attention mechanism, the improvement is relatively high; the increase is 2.1 points. In the number 2 model, we added a Gaussian noise layer and this model reached an F1-score of 0.8438; the increase is 2.18 points.

We found that the best standard deviation in Model No. 1 is 0.1; the perturbations must be small for the best performance. In contrast with Model No. 2, the best value is 0.3 (see Table 10).

The PW-CNN model not only captured better details, but also predicted DDI relations. The MCCNN model, through multichannel embedding, expressed terms with greater precision and richness. Figure 8 shows the overview of the evaluation of the models with respect to the number of examples. It is observed that Model No. 2 based on RBERT architecture performs better than the others, with different numbers of examples from the DDI dataset. It performs well, even with few training data.

The final results are summarized in Table 11. They are divided by linear methods, kernel methods and neural networks. The models based on neural networks belong to recent studies based on deep learning techniques for the DDI classification task.

## 4. Discussion

When more data are added to any ER model, the number of error cases decreases, as observed in the experimentation and results with the two chosen models. As mentioned previously, the Gaussian noise technique was used for data augmentation.

Next, we will present a comparison of the most outstanding models in this challenge and contrast them with our models and we will perform an error analysis using the confusion matrix.

### 4.1. State-of-the-Art Method Performance Comparison

In this section, we conduct a comparative analysis of our approach with existing methods that were previously developed for DDI extraction. We evaluate these benchmark methods using the same training and test dataset that was provided by the DDIExtraction2013 challenge. These methods can be broadly classified into two categories: traditional approaches and those based on neural networks.

The following are the traditional methods: UTurku [16], WBI [39], FBK-irst [17], Kim [37], UWM [36] and Nil [38].

In this first category, characteristic-based methods were used for DDI classification. UTurku was adapted from the TEES Turku Event Extraction System. To extract the main characteristics, TEES employs a combination of domain-dependent resources and dependency analysis. On the other hand, the two-stage methods of UWM and Kim use SVM models with contextual, lexical, semantic and tree-structured features in both stages.

Moreover, kernel-based methods provide powerful techniques for using graph-based functions. WBI and FBK-irst are two-stage approaches to DDI classification. In the first stage of these models, different kernel methods were used that were based on characteristics of dependency trees and syntax. Then, in the second stage, the WBIs used TEES and the FBK-irst used an SVM with a non-linear core for classification. At Nil, they used a multiclass SVM as a one-stage kernel method.

In the second category, methods based on deep neural networks were used. These have emerged as promising approaches to machine learning of functions and have become a dominant method for the DDI-extraction task. Convolutional neural networks can learn local characteristics through discrete convolution with different filter sizes.

Improving the performance of DDI extraction is currently imperative as it is measured by the F1-score metric. The primary difficulty is precisely detecting and classifying DDIs in complex biomedical phrases. For instance, the longest sentence in the DDIExtraction2013 corpus contains more than 150 words, making it a daunting task for deep neural networks. Recent research has shown that dividing a sentence into multiple parts based on the entities present can effectively enhance the performance of relation extraction.

The model of a BI-LSTM can retain information about past and next words while reading. Therefore, when we apply grouping in the output of BI-LSTM, we can obtain functions that contain information about the complete context of the complete sentence. The work by Zhang et al. [34] presents a hierarchical model of a recurrent neural network that integrates SDP, which is the shortest dependency path, for DDI-type ER. It reaches an F1-score of 0.729; this study is one of the most referenced in this field.

Liu et al. [6] propose a method based on a basic convolutional neural network (CNN) that reaches an F1-score of 0.698. The work of Quan et al. [18] proposes an MCCNN model based on a CNN type, which uses multiple channels to represent each word in sentences. Inspired by the idea of using three parallel channels to classify an image (RGB) with a CNN, the model integrates multiple embeddings of words from different sources to increase the semantic load of each word, PubMed, PMC, MedLine and Wikipedia [41], and another inclusion of words using MedLine with the application of the Continuous Bag of Words (CBOW) model.

Park et al. [20] use a PW-CNN model that automates the extraction process and captures structural information between drug entities. Zeng et al. [42] propose a PW-CNN model with a convolution by parts; therefore, it divides the sentence containing a pair of detected entities into three parts, taking advantage of the location information of the entity to retain the structural characteristics of a sentence. Subsequently, it applies the max-pooling technique and, finally, the Softmax classifier. Additionally, it makes use of five versions of word embeddings as PW-CNN channels to enrich unknown terminology.

Our Model No. 1 was implemented based on the work of Park et al. [20], which makes use of robust strategies to face this challenge; an attention mechanism was added to give more relevance to important words within the sentence and additionally a data augmentation technique is used in the area of computer vision to improve the performance of the model, Gaussian noise, which inserts perturbations at the output grouping, with which it was possible to improve the F1-score metric, as shown.

BioBERT (Model No. 2) was implemented based on the work of Nguyen and Ho [35]. This approach uses the Relation Bidirectional Encoder Representations from Transformers (RBERT) architecture to classify DDIs from biomedical texts. BioBERT significantly outperforms state-of-the-art models on the following representative biomedical text-mining tasks: biomedical named entity recognition and biomedical relation extraction. Pre-training BERT on biomedical corpora improves its understanding of complex biomedical texts, as evidenced by the results. By way of the insertion of a Gaussian noise layer, this model improved performance, which indicates that the use of data augmentation techniques provides support with respect to improving the performance of these models.

### 4.2. Error Analysis

As mentioned earlier, due to the complexity of biomedical text, extracting relations from this corpus remain a huge challenge. In this section, we analyze the errors caused by MCPCNNGauss and BERTGauss models. In the same manner as the works of Zhu et al. [43], Huang et al. [44], we showed a confusion matrix for error analysis; see Figure 9 and Figure 10. The color concentration corresponds to the proportion of the relation.

Figure 9 illustrates the findings of the misclassified proportion clearly and lists the common errors as follows:The instances of Negative type are more confused with instances of Mechanism and Effect type than with the Advice and Int type. Injecting Gaussian noise improves the prediction except in the Int type.We observed that the instances of Mechanism type are more confused with instance of Negative type than with the Effect, Advice and Int type. Adding Gaussian noise there is much lower error; however, in the Int type instance the error is 0.We obtained that all the instances of Effect type are more confused with instances of Negative type than with the Mechanism, Advice and Int type. As mentioned earlier, with Gaussian noise there is much lower error, but in the Int type instance the error is 0.It works on the same principle as earlier, all the instances of Advice type are more confused with instances of Negative type than with the Mechanism, Effect and Int types. There is a difference with Gaussian noise; there is much lower error but in the Mechanism type instance the error is 0; this means there is no misclassification.We note that the instances of Int type are easily confused with the instances of Effect type. There is no misclassification in the instances of Advice type. The results indicate that there is not much difference when we insert Gaussian noise.

Comparing Figure 9 and Figure 10 shows that the BERT model is more powerful than the MCPCNN model. We explain the most common error in the MCPCNN model:A considerable number of the instances of Negative type are more confused with instances of Mechanism, Advice and Effect type than with Int type. Injecting Gaussian noise improves the classifier prediction.The accuracy percentage in this model is lower than the BERT model. Good results were obtained in the first “Negative” instance and reasonable results in the others. Similar behavior was observed in the Mechanism, Effect, Advice and Int instances as in the BERT model. The addition of Gaussian noise improves the performance of the model, especially in the instances of Negative type.

As can be seen in Figure 9 and Figure 10, The initial mistake highlights the classifier’s inability to distinguish between positive and negative samples, as some positive samples are misclassified as negative due to an imbalanced dataset with significantly more negative training data than positive. Generally, the data imbalance could be considered as the reason for this error. The second error may be attributed to the classifier’s misclassifications of instances of Int type, as the small sample size of only 188 instances could limit the classifier’s ability to recognize this type of instance accurately. In the third error, negative samples are also incorrectly classified as instances of Effect type, implying that instances of Effect type are characterized by complex semantics and structures. The last error confuses several Mechanism, Advice, Effect and Int samples with Negative samples, and shows semantic similarity.

We found evidence to suggest that our models misclassified sentences such as the following:“Otherype of type drugs which may s of enhancetypethe neuromuscular blocking action of nondepolarizing agents such as <e1> MIVACRON </e1> include certain antibiotics (e.g., aminoglycosides, tetracyclines, bacitracin, polymyxins, of Int typeclindamycin, colistin and sodium colistimethate ), magnesium salts, lithium, local anestheticsprocainamide of, type and <e2> quinidine </e2>”.“<e1>Barbituratessof type </e1> may decrease the effectiveness of oral <e2>sofcontraceptivestype</e2>, certain antibiotics, quinidine, theophylline, corticosteroids, anticoagulants and beta blockers”.

The terms “increase” and “decrease efficacy” can be identified in these sentences; they may confuse models with an Effect relation. These findings indicate a possible semantic similarity between Effect and Int types of sentences, as well as the complexity of the semantics and structures associated with Effect relations, demonstrated by misclassifications of negative samples.

## 5. Conclusions

A few prior works have documented the effectiveness of an approach to text data augmentation. In this study, we tested Gaussian noise perturbations. We found that when a Gaussian noise layer is added to these models the performance of deep learning techniques improves with respect to the task of extracting biomedical relations, particularly for situations with a limited number of available annotations, which is the case of the Medline dataset of the DDIExtraction2013 challenge.

On the other hand, data augmentation techniques are being used widely in the field of computer vision and recently these techniques have been used in relation-extraction tasks in natural language processing (NLP).

In this study, we tested how to reduce the number of misclassifications of two models that have excellent performance by the use of the data augmentation approach using Gaussian noise perturbations.

We found that the error analysis clarifies the details of the misclassification since the unbalanced data can affect the performance of the model. These results show that by adding a layer of Gaussian noise, it is possible to predict more accurately the relation of type Int, which is the most complex class and which easily confuses the relations of type Negative, Effect, Mechanism and Advice and this is the reason why previous models do not achieve high scores.

Our results provide compelling evidence for the classification part, most notably in the BioBERTGauss model where can observe that no effect sample and no mechanism sample is misclassified into the relations of type Int.

This study therefore indicates the benefits gained from our BERT model, which encompasses the language model BioBERT which has been comprehensively pre-trained on the biomedical corpus. BioBERT can learn the semantic representation of the input sentences and entities.

As can be seen, the Gaussian noise injection gave satisfactory results for our two models. Our study will provide the accuracy framework for future studies to evaluate data augmentation techniques and improve NLP models. Future works are mentioned below:In the next stage, the Gauss model is replicated for BIOALBERT (A Lite Bidirectional Encoder Representations from Transformers for Biomedical Text Mining); it generates very good results in the biomedical task NLP.Future works should apply other data augmentation approaches used in the area of computer vision that are successful in visual tasks.There is a wide range of biomedical text corpora that fit well within our framework, but their complexity would require further research.We expect we will provide insights into the development of generalizable NLP for clinical text models and task relation extraction that could lead to improvements in models.

## Figures and Tables

**Figure 1 pharmaceutics-15-01823-f001:**
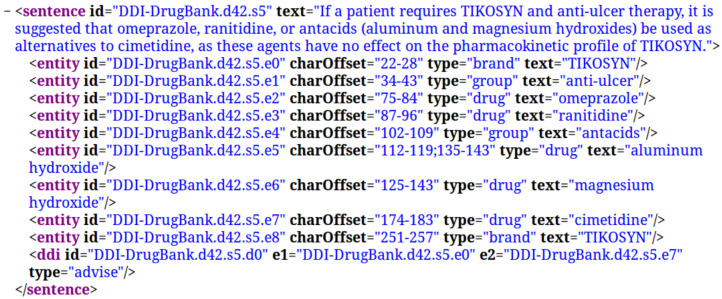
Annotated corpus example.

**Figure 2 pharmaceutics-15-01823-f002:**
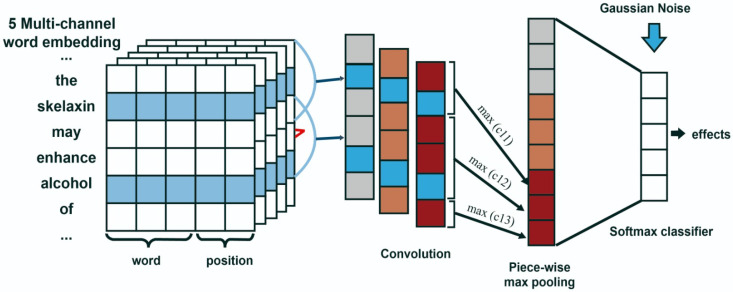
Architecture of the MCPCNNGaus model.

**Figure 3 pharmaceutics-15-01823-f003:**
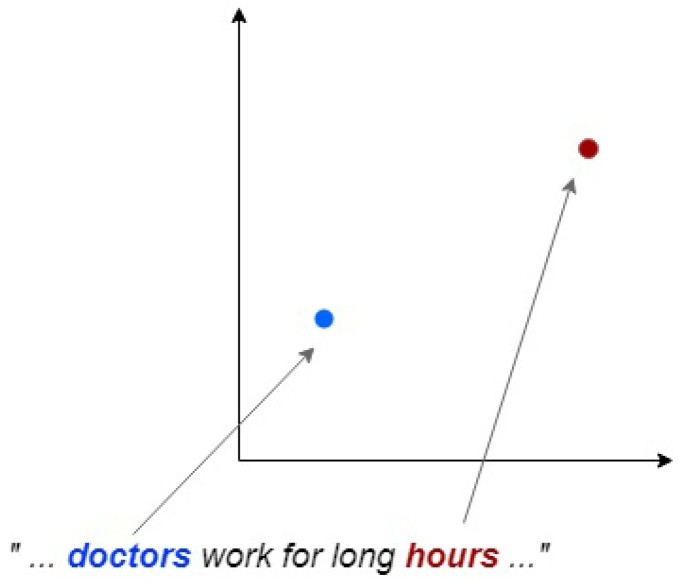
Each word in a sentence is mapped.

**Figure 4 pharmaceutics-15-01823-f004:**
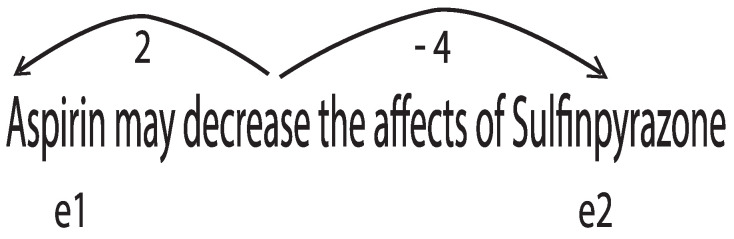
Word position in the input sequence [20].

**Figure 5 pharmaceutics-15-01823-f005:**
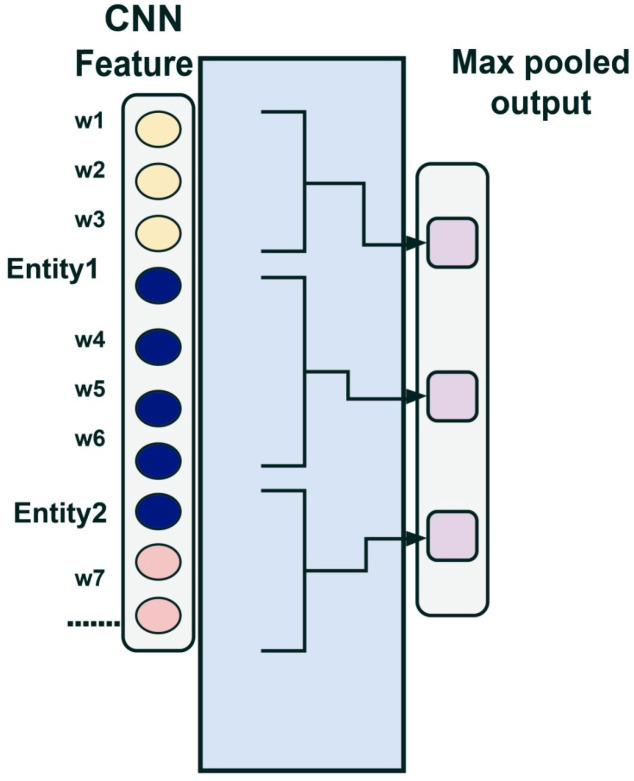
PW-CNN Architecture Model.

**Figure 6 pharmaceutics-15-01823-f006:**
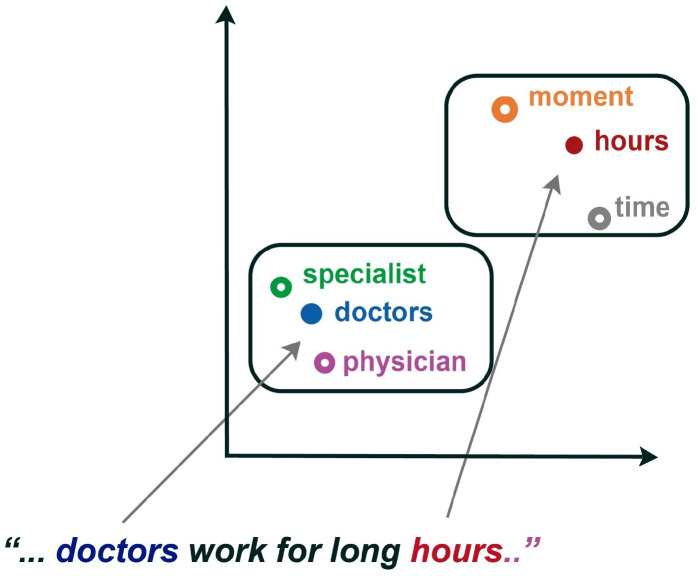
Perturbations are performed based on nearest neighbors.

**Figure 7 pharmaceutics-15-01823-f007:**
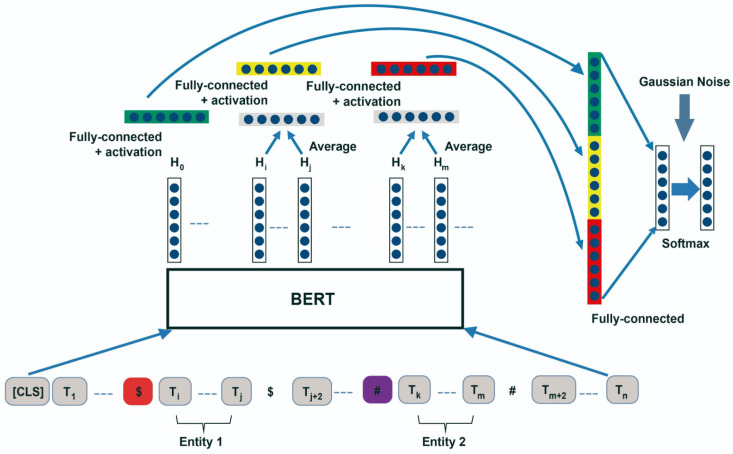
Architecture of Pre-trained Model BERT.

**Figure 8 pharmaceutics-15-01823-f008:**
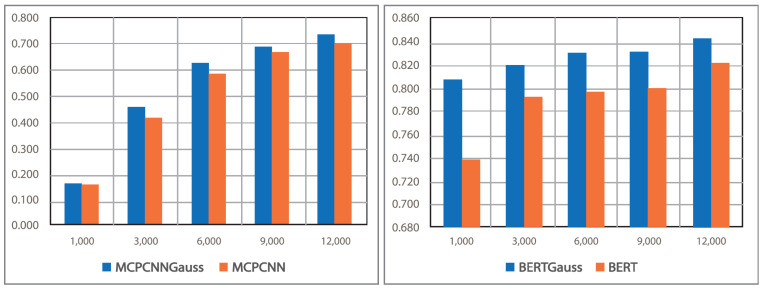
Model evaluation, in accordance with the number of examples.

**Figure 9 pharmaceutics-15-01823-f009:**
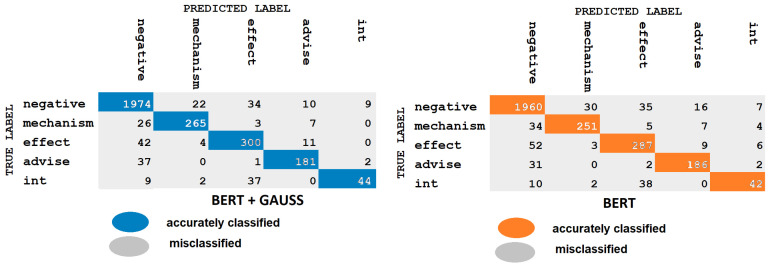
Confusion matrix BERT.

**Figure 10 pharmaceutics-15-01823-f010:**
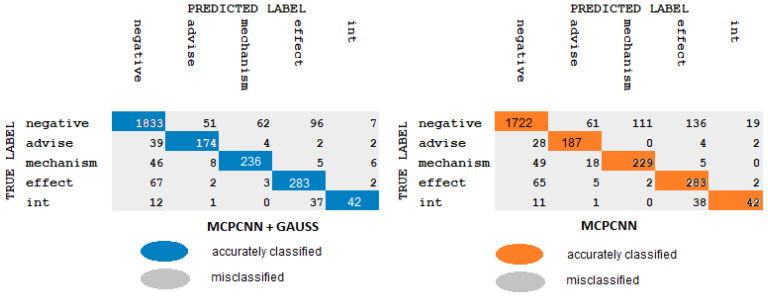
Confusion matrix MCPCNN.

**Table 1 pharmaceutics-15-01823-t001:** Some examples showing several types of drug–drug interactions.

No	Sentence	Class
1	“Fluoxetine” and crocin should not be administered to patients receiving “phenelzine”	Advice (Fluoxetine, Phenelzine)
1	Fluoxetine and “crocin” should not be administered to patients receiving “phenelzine”	Advice (Crocin, Phenelzine)
2	Both “PGF2alpha” and “Oxytocin” induced dopamine release in the nucleus accumbens	Effect (PGF2alpha, Oxytocin)
3	The half-life of “ketamine” in plasma and brain was longer in the presence of “halothane”	Mechanism (Ketamine, Halothane)
4	The drug interaction between “warfarin” and “rifampin” is not well known	Int (Warfarin, Rifampin)

**Table 2 pharmaceutics-15-01823-t002:** DDIExtraction2013 Corpus.

Class	Train: DrugBank	Train: Medline	Total	Test: DrugBank	Test: Medline	Total
Document	572	142	714	158	33	191
Advice	818	8	826	214	33	221
Effect	1535	152	1687	298	33	331
Mechanism	1257	62	1319	278	33	311
Int	178	10	188	94	2	96

**Table 3 pharmaceutics-15-01823-t003:** Statistics for five word-embedding models (all with 200 dimensions).

Training Corpus	Number of Words
PMC	2,515,686
PubMed	2,351,706
PMC and PubMed	4,087,446
Wikipedia and PubMed	5,443,656
Medline	650,187

**Table 4 pharmaceutics-15-01823-t004:** Generated Instances.

Pair of Drugs ($e_{1}$,$e_{2}$)	DDI Candidates Are Anonymized
(Aminoglutethimide, coumarin)	Aminoglutethimide $drug_{1}$ decreases the effect of coumarin $drug_{2}$ and warfarin $drug_{n}$.
(Aminoglutethimide, warfarin)	Aminoglutethimide $drug_{1}$ decreases the effect of coumarin $drug_{n}$ and warfarin $drug_{2}$.
(coumarin, warfarin)	Aminoglutethimide $drug_{n}$ decreases the effect of coumarin $drug_{1}$ and warfarin $drug_{2}$.

**Table 5 pharmaceutics-15-01823-t005:** Model 1: Settings.

Parameter	Select
Batch size	64
Word embedding size	200
Kernel CNN size	[3 5 7 9]
Number of filters	128
Dropout rate	0.45
Adam learning rate	3.00 × 10${}^{-4}$
Gaussian noise	0.10

**Table 6 pharmaceutics-15-01823-t006:** Model 2: Settings.

Parameter	Unit
Batch size	16
Optimization step	4015
Max position embeddings	512
Model type	BERT
No. hidden layers	12
Dropout rate	0.1
Vocab size	28,996
Gaussian noise	0.3

**Table 7 pharmaceutics-15-01823-t007:** Comparing the effects of pretrained language models.

	F1
BERT-based-cased	0.806
SciBERT-scivocab-uncased	0.812
BioBERT v1.0 PubMed PMC	0.818
BioBERT v1.1 PubMed	** 0.822 **

**Table 8 pharmaceutics-15-01823-t008:** Model Evaluation.

Model	Precision	Recall	F1-Score
BI-LSTM	0.6817	0.6421	0.6516
PW-CNN	0.7159	0.654	0.6835
MCCNN	0.7242	0.6787	0.707
* MCPCNN	0.6895	0.7477	0.7174
* BioBERT	-	-	0.822

**Table 9 pharmaceutics-15-01823-t009:** Strategy effect and the impact on performance.

Model	F1-Score
MCPCNN	0.717
MCPCNN + Gauss	0.738
BioBERT	0.822
BioBERT + Gauss	0.8438

**Table 10 pharmaceutics-15-01823-t010:** Noise Gaussian—Standard deviation.

Noise	MCPCNN + Gauss	BioBERT + Gauss
0.1	0.738	0.829
0.2	0.712	0.830
0.3	0.7016	0.8438

**Table 11 pharmaceutics-15-01823-t011:** Performance comparison between the proposed methods.

Method Class	Method	Precision	Recall	F1-Score
Linear methods	UTurku [16]	0.732	0.499	0.594
	UWM [36]	0.439	0.505	0.47
	Kim [37]	-	-	0.67
Kernel methods	Nil [38]	0.535	0.501	0.517
	WBI [39]	0.642	0.579	0.609
	FBK-irst [17]	-	-	0.67
RNN Methods	Liu et al. [6]	0.757	0.647	0.698
	Quan et al. [18]	0.76	0.653	0.702
	Park et al. [20]	0.6895	0.7477	0.7174
	Zhang et al. [34]	0.741	0.718	0.729
	Wuti et al. [40]	0.801	0.740	0.770
This work	Proposed Model No. 1	0.7359	0.7405	0.7382
	Proposed Model No. 2	0.837	0.85	0.8438

## Data Availability

Not applicable.

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
