# Peer review of "Improving Drug–Drug Interaction Extraction with Gaussian Noise"

_pharmaceutics, 2023, doi:10.3390/pharmaceutics15071823_

Round 1

Reviewer 1 Report

  Drug-drug interactions are important not only for rational clinical use, but also for post-marketing pharmacovigilance. To address this problem, the detection and extraction of drug-drug interaction information from the literature has become a research topic in biology, medicine, pharmacology and bioinformatics in recent years. In this manuscript, authors develop two models to extract drug-drug interaction information based on the Gaussian noise and pre-trained BERT language model. The performance of the current method has been evaluated and confirmed through comparison with literature methods based on the benchmark datasets. The following issues need to be addressed.

1.        Line 113 and 114. The DDI 2013 corpus contains five DDI types: Advice, Effect, Mechanism, Int and false. However, in Table 2, Class: Document, Advice, Effect, Mechanism and Int. Document is false?

2.        As far as I know, precision and recall are commonly used to evaluate the performance of binary classification models. This current study falls under the category of five classification problems, how to calculate these metrics?

Author Response

I take into account your suggestions and comments, which I have considered carefully. In my response, I hope to address each of the points you have raised clearly and concisely. Likewise, I hope that this response is constructive and contributes to improving the quality of my work.

  1. Only positive instances are relevant, which can be divided into four types: Effect, Mechanism, Advice, and Int. There is a large number of negative instances (pairs of drugs that do not interact with the context), which typically affect the performance of DDI extraction systems based on machine learning due to the data imbalance problem. Therefore, filtering as many negative instances as possible is very important for the subsequent DDI extraction module. In the preprocessing phase, these instances are filtered based on the following criteria: (1) the two drugs have the same name, (2) one drug is an abbreviation or acronym of the other, (3) the two drugs appear in the same coordinate structure that has more than two drugs as elements, (4) one drug is a special case of the other.
  2. Most DDI extraction methods use F-score, precision and recall as evaluation metrics, as these metrics are widely used in the field of machine learning and natural language processing to evaluate the performance of binary and multiclass classification models.

    The F-score is a harmonic mean of precision and recall, which provides a single score that balances both metrics and is especially useful when classes are unbalanced or imbalanced as is the case in our corpus. These metrics provide information on the ability of the classification model to identify true positives, false positives and false negatives, which is paramount for identifying potential drug-drug interactions.

    Using these metrics to evaluate the performance of DDI extraction methods allows researchers to compare different models and techniques and identify the strengths and weaknesses of each approach. In addition, these metrics allow researchers to optimize their models and algorithms to achieve higher accuracy and recovery rates in detecting potential drug-drug interactions.

Reviewer 2 Report

The study focuses on a method for improving drug-drug interaction searches by exploting Gaussian noise. The method is applied to the Piecewise Convolutional Neural Network (PCNN) and the Bidirectional Encoder Representations from Transformers (BERT) models.
It is not clear what the novelty of the method is. The PCNN and BERT language models are well-documented tools, and there is an extensive literature devoted to these models. Application of Gaussian distribution to enhance representation or improve performance of the models is also a well-known trick that is widely used in numerous studies. For example, it was used for the PCNN model in Applied Intelligence (2022) 52: 4599–4609 and within the BERT approach in IEEE Journal of Biomedical and Health Informatics 26.3 (2021): 1341–1352. Moreover, it was effectively used in a wide context of drug-protein interactions searches (see, for example, Bioinformatics 36.15 (2020): 4323–4330, etc).

Without a description of the connections between the suggested improvements and the earlier studies, the manuscript cannot be published.
A quick explanation of how the gaussian distribution method has been applied to enhance the search would be helpful.
I also suggest changing the content, moving the majority of well-known material (such as descriptions of PCNN and BERT models) to Supporting Information and putting more of an emphasis on the innovative aspects of the model and the specifics of how it is used.

Author Response

In our study, a data augmentation technique based on Gaussian noise is implemented to produce more examples. For this purpose, a Gaussian noise layer is implemented before applying the softmax activation function in both proposed models. The data augmentation is generated based on perturbations.

In the article "PCNN in Applied Intelligence (2022) 52: 4599-4609," the Gaussian distribution technique is used to model the relationship between words with and without entities to assign weights to the words in the sentence, thereby reducing the influence of noisy words. In the other study, "Chemical-protein Interaction Extraction via Gaussian Probability Distribution and External Biomedical Knowledge," the Gaussian probability distribution is used to capture the local structure of the instance and improve performance.

I find the suggestions proposed to be correct, and I will implement them.

Reviewer 3 Report

Authors have presented improvements in methodological approach to studying drug-drug interactions in an existing dataset.

Since the original dataset is pretty outdated, I would suggest authors to add some insights into how a new dataset could be prepared and how it could be analyzed using the proposed methodology. More specifically, in the context of explainable AI.

To what extent is the model useful for extraction, and to what extent to prediction of drug-drug interaction. How well would it perform on a new dataset?

Author Response

The DDI extraction 2013 corpus was developed by a team of researchers from the Biomedical Informatics Laboratory at the University of Navarra in Spain. The team was led by Dr. Isabel Segura-Bedmar and Dr. Paloma Martínez, and involved the collaboration of other researchers and experts in the fields of pharmacology and biomedical informatics. The corpus was developed to facilitate the advancement of research in the extraction of drug-drug interactions through the use of natural language processing techniques and machine learning. Since then, it has been widely used by researchers worldwide in various projects related to computational pharmacology and biomedical informatics.

In addition, it is one of the largest and most complete corpora available for this field of research, making it very useful for the training and evaluation of machine learning models and natural language processing techniques. Nowadays, it is widely used in scientific research and in the annual competition of drug-drug interaction extraction (DDIExtraction 2013) to evaluate the performance of different approaches and systems. In summary, the DDI extraction 2013 corpus is a popular and robust option for the extraction of pharmacological interactions due to its quality, size, and relevance to the field.

One possible corpus that could be used is the BioNLP-ST 2016, which extracts relationships between proteins, molecules, and diseases. It would be interesting to consider using this framework in future work to see the results obtained with this corpus.

Round 2

Reviewer 2 Report

Agree with corrections and author explanations